# MicroRNA-199a-5p attenuates blood-brain barrier disruption following ischemic stroke by regulating PI3K/Akt signaling pathway

Guangxiao Ni[1]*, Lulu Kou[1], Chunqiao Duan[1], Ran Meng[1], Pu Wang[2]

1 Department of Rehabilitation of the Second Hospital of Hebei Medical University, Shijiazhuang, China,
2 Stomatological Laboratory of the Second Hospital of Hebei Medical University, Shijiazhuang, China

* ngx7912156@163.com

## Abstract

### Objective

To explore whether miR-199a-5p regulated BBB integrity through PI3K/Akt pathway after ischemia stroke.

### Methods

Adult male Sprague-Dawley rats with permanent middle cerebral artery occlusion(MCAO) were used in experiment. The Ludmila Belayev 12-point scoring was used to measure the neurological function of MCAO rats. The Evans Blue Stain, immunofluorescence staining, western-blotting and RT-PCR were performed to evaluate the effects of miR-199a-5p mimic on BBB integrity in rats following MCAO.

### Results

The result suggested that miR-199a-5p mimic treatment possessed the potential to boost proprioception and motor activity of MCAO rats. MiR-199a-5p decreased the expression of PIK3R2 after MCAO, activated Akt signaling pathway, and increased the expression of Claudin-5 and VEGF in the ischemic penumbra. Furthermore, miR-199a-5p alleviated inflammation after cerebral ischemia. BBB leakage and neurocyte apoptosis were cut down in MCAO rats treated with miR-199a-5p mimic.

### Conclusions

MiR-199a-5p mimic decreased the expression of PIK3R2 and activated Akt signaling pathway after ischemia stroke, reduced the expression of inflammatory cytokines, and attenuated BBB disruption after ischemic stroke.

**Data Availability Statement:** All relevant data are within the manuscript and its Supporting Information files.

**Funding:** National Natural Science Foundation of China, (No.81273609) Natural Science Foundation of Hebei Province, (No. H2023206115).

**Competing interests:** All authors declare that they have no conflicts of interest to report.

## Introduction

Stroke has been the major risk threatening human health. China has become the region with the highest incidence rate and prevalence of stroke in the world. However, ischemic stroke accounts for approximately 60% -85% [1–3]. After stroke, disability severely reduced people's quality of life [4,5]. following ischemic stroke, the expression of various inflammatory cytokines increased, leading to damage to the integrity of the blood-brain barrier (BBB) [6,7]. When cerebral ischemia occurred, the permeability of the BBB increased, harmful components in blood vessels seeped into brain tissue, exacerbating cerebral injury [5,8,9].

The disruption and dysfunction of the BBB leading to leukocyte infiltration, cerebral edema, and hemorrhagic transformation were common mechanisms [10]. Maintaining the integrity and normal function of the BBB to protect the central nervous system(CNS) from secondary damage was a potential therapeutic strategy for ischemic stroke [11]. The BBB was composed of endothelial cells, pericytes, astrocyte terminals, and basement membrane, regulated substance exchange between the CNS and peripheral blood circulation. The BBB played a key role in the homeostasis of the CNS [10,12]. Multiple factors were involved in regulating the structure and function of the BBB. Especially, the continuous tight junctions (TJs) between cells, which were the main structure ensured selective permeability of the BBB [10,13]. TJs were closed structures formed by the fusion of specific transmembrane proteins from adjacent cell membrane outer layers, located between brain microvascular endothelial cells [14]. TJs were mainly composed of various proteins such as transmembrane proteins [Claudin, Occludin], cytoplasmic attachment proteins, cytoskeletal proteins, and other proteins that regulated BBB permeability [15]. Particularly, claudin-5 was the predominant TJs protein that influenced the selective permeability of the BBB, and inflammation induced its downregulation and BBB disruption [11,15,16].

MicroRNA (miRNA) was a kind of endogenous non coding RNA with length about 19-25nt, which was widely involved in the post transcriptional regulation of genes. The mechanism of miRNA was to inhibit protein translation and induce target RNA degradation through incomplete miRNA mRNA complementary pairing. miRNA has become potential clinical biomarker for diagnosis and prognosis of many diseases [17–20]. MiR-199a-5p was widely expressed in brain tissues, such as the hippocampus, olfactory bulb, subventricular zone, cortex, and striatum, and played an important role in physiological regulation processes such as neurogenesis and plasticity [21,22]. MiR-199a-5p prevented hypoxia induced injury of brain spinal cord and myocardium [3]. MiR-199a-5p promoted endogenous neurogenesis, improved neurological function after ischemic stroke, and reduce infarct volume [3,23,24].

Compared with healthy people, the expression of miR-199a-5p in the serum of patients with cerebral ischemia was obviously declined, which indicated that miR-199a-5p was essential in the process of cerebral ischemia [24,25]. MiR-199a-5p participated in regulating angiogenesis and vascular integrity by directly inhibiting the phosphatidylinositol-3 kinase regulatory subunit 2 (PIK3R2) [26], which was a negative regulator of the vascular endothelial growth factor (VEGF) pathway [26,27]. The expression of PIK3R2 had intimately correlation with Akt pathway [27,28].

Studies indicated that miR-199a-5p improved cognitive function and reduced hippocampal neuronal apoptosis in ischemic stroke rats by regulating the AKT signaling pathway [19]. These findings suggested that miR-199a-5p probably protected BBB integrity and reducing neuronal apoptosis after cerebral ischemia. Therefore, we hypothesize that miR-199a-5p could maintain BBB integrity and reduce neuronal apoptosis after ischemic stroke by regulating the Akt signaling pathway. The present study used the middle cerebral artery occlusion rats to explored the hypothesis.

## Materials and methods

### Ethics statement

All animal experiments were performed in strict accordance with the Guidelines for Animal Experiments at the second hospital of Hebei Medical University. The experimental protocols were approved by the Animal Experiments Committee of the second hospital of Hebei Medical University (Permit number: 2023-AE-200). All procedures were performed under isoflurane inhalation anesthesia, and every effort was made to minimize the suffering of the rats.

### Animal and experimental groups

Adult male Sprague-Dawley(SD) rats (260–280g) used for the experiment was purchased from Weitong Lihua Laboratory Animal Technology (Beijing, China). The rats lived in cages which filled with sterile wood shavings as bedding material under 12/12 h light/dark cycle at 22˚C± 3˚C and 55% ± 10% relative humidity. Rats had free access to food and water. The experiment started after 7 days of adaptive feeding.

The SD rats were randomly divided into four groups: Sham group, MCAO group, miR-199a-5p group, miRNA-NC group.

### Modeling and intervention of rats

Using middle cerebral artery occlusion (MCAO) was applied to induce ischemic stroke. The rats were anesthetized with 3.5% isoflurane. Following anesthesia, the right middle cerebral arteries of rats were occluded by way of inserting a 6–0 nylon monofilament (depth about18.5 ± 0.5mm). Laser Doppler blood flow measurement successfully ensured occlusion (MCA blood flow reduction exceeded 75%). Ligated the remaining end of the ECA, cut the redundant part of the nylon, and sutured the incision. After MCAO, temperature of the rats was maintained at 36.5–37˚C. The rats in the Sham group were performed without any injury of arteries.

In the light of the Zea Longa Behavior Rating Scale, the neurological function was divided into five points: 0 indicated no neurological deficits; 1 point was lightly focal neurological deficits (the left front paw failed to fully extend); 2 points was classified as medium focal neurological deficits (leftward rotation); 3 points were serious focal deficits (tilted to the left); 4 points was unable to walk and had an inferior consciousness. Excluded MCAO rats with scores 0 and 4, and supplied in time.

Mixed separately miR-199a-5p mimic and negative control (Gemma Gene, Shanghai) with pure water(250μl) and dissolved. Transfection reagent Enterster TM in vivo (Ingen, Beijing) diluted with physiological saline, and 33μl Added 217μl physiological saline. The diluted in vivo transfection reagent mixed with microRNA yielded 10μl microRNA solution. The rats were fixed on a stereotactic frame. The solution was injected slowly into the right lateral ventricle of rats for 5 min (0.8mm posterior to bregma, 1.5mm lateral to midline, 3.5mm below the surface of the skull). In miR-199a-5p group and miR-NC group, rats were injected separately with miR-199a-5p mimic(5μl) and miR-NC(5μl). Injected equal volume of transfection agent into right lateral ventricle of rats in Sham group and MCAO group. After injection, left the needle with the syringe for another 2 minutes. Then removed the needle and sealed the hole with bone wax, sutured the skin, and immediately perform MCAO.

### Neurobehavior assessment

Ludmila Belayev 12-point scoring (LB 12 Scoring) was applied to assess the sensorimotor integration function of rats 24 hours after MCAO. LB 12 Scoring included posture reflex test and

limb placement test. The postural reflex test was a tail lift suspension test: 0 points for no neurological deficits, 1 point for normal limb flexion, and 2 points for positive pushing test. The limb placement test included: firstly, in the visual test, the researcher held the rat in his hand and suspended its forepaws. Placed the rat 10 centimeters over the table and approached the table slowly. The natural reaction of rats was to immediately grab the table with forelimbs, while MCAO rats moved slowly or could not accurately touch the table. 0 points: Normal response in rats; 1 point: Slow reaction, but not exceeding 2 seconds; 2 points: Slow reaction for more than 2 seconds. Horizontal stimulation, placed the rat above the desktop side, and followed the same procedure and scoring criteria as before. Tactile test included front and side stimuli. The rat's eyes were entirely covered and its forepaws suspended in the air. The skin and hair of their forepaws gently touched the table. The response and scoring criteria of rats were identical with visual tests. Thirdly, the proprioception test, only involved anterior stimulation, manipulation and scoring criteria were identical with tactile test, which evaluated the sense of space, position and balance of rats. LB 12 Scoring ranged from 0 to 12 points. The more serious, the higher the score.

## Tissue preparation

To avoid causing distress, the rats were euthanized with an overdose 3.5% isoflurane at 24 hours after neurobehavior test. Eye vein blood of rats were collected and serum was extracted, stored at -80˚C. 0.1M phosphate buffer solution (PBS, pH 7.4) was perfused into through left ventricle. Subsequently, 4% paraformaldehyde was perfused for fixation. The right brain tissues were removed and separated infarct area and ischemic penumbra area, and then stored at -80˚C.

## Quantitative real-time PCR

Frozen serum and brain tissue were homogenized using ultrasound in 1ml triazole reagent. Incubated the homogenate at 25˚C for 5 minutes to isolate the complex nucleoprotein. Then, added 0.2ml chloroform, shook the homogenate and incubated at 25˚C for 5 minutes. Next, centrifuged the sample at 10000g at 4˚C for 15 minutes. Mixed the aqueous phase including RNA with 0.5ml isopropanol, incubated at 25˚C for 10 minutes, and centrifuged at 10000g at 4˚C for 10 minutes. Resuspended RNA particles in 75% ethanol, centrifuged under 3500g at 4˚C for 5 minutes, dried and dissolved in 20μl 0.1% DEPC water. 30 μl RNase-free ddH2O was added at 25˚C for 2 minutes. Dissolved the precipitate thoroughly and obtained the total RNA of the sample, and then extracted it using Trizol reagent (Invitrogen Corp China).

Synthesized cDNA in the light of the M-MLV conserved transcriptase specification (Invitrogen Corp China). PCR reaction was performed using 1.5ml cDNA sample and SYBR Primex Ex Taq (TaKaRa, Kusatsu, Japan) in 20ml by the Exicycler 96 system (Bioneer, Daejeon, Korea). Calculated the relative expression by comparative $2^{-\triangle\triangle CT}$. All procedures were performed separately more than 3 times. The primers for RT-PCR were showed in Table 1.

**Table 1. The primer information.**

| Gene | Forward Primer | Reverse Primer |
|---|---|---|
| miR-199a-5p | 5'-GCATCGTCGTACCGTGAGTAAT-3' | 5'-GTGCAGGGTCCGAGGTATTC-3' |
| Claudin-5 | 5'-TGGTGCTGTGTCTGGTAGGATGGA-3' | 5'-GTCACGATGTTGTGGTCCAGGAAG-3' |
| VEGF | 5'-TCACCAAGGCCAGCACATAG-3' | 5'-GGGCACCAACGTACACGC-3' |
| β-actin | 5'-GGCACCCAGCACAATGAA-3' | 5'-AGAAGCATTTGCGGTGG-3' |

## Western blotting

Expression levels of Claudin-5, VEGF, PIK3R2, P-AKT, and AKT in cerebral ischemic penumbra were measured by Western blotting. Homogenized the tissue in lysis buffer, which contained 10 μl inhibitor mixture in 500 ml RIPA buffer. After boiling with SDS for 5 minutes, the segregated proteins were electrophoretically on 8% polyacrylamide and then transferred to PVDF membrane. Incubated non-specific binding sites overnight in TBST (20mM Tris HCl, pH 7.61mM NaCl, 0.05%) at 4°C with anti Claudin-5 (1:500, Abbkine, USA), anti VEGF (1:500), anti PIK3R2 (1:1000, Boster, China), anti P-AKT (1:1000, Abcam, China), and anti AKT (1:500, Santa Cruz, USA). Washed 3 times with TBST, and incubated by goat anti-rabbit IgG horseradish peroxidase (HRP)-conjugated secondary antibody (Abcam, China) for 60 minutes. Immunoblotting was washed 3 times with TBST and emerged on X-ray film. Protein molecules size was decided through moving the protein ladder (Fermentas, Canada) in adjacent lanes. Image J software was used to scan and quantify film signals. Anti β-Actin (1:1000, Abcam, China) was an internal control and the relative protein levels were standardized to β-Actin.

## Evans blue (EB) test of Blood brain barrier (BBB) permeability experiment

After neurological function assessment, 2% Evans blue dye (2mL/kg) was injected through the tail vein. After 2 hours, the rats were euthanized and treated with physiological saline from the left ventricle. The brain was excised, weighed, incubated at 37°C for 24 hours, centrifuged at 2000g for 10 minutes, collected the supernatant. Measured the absorbance value at a wavelength of 632 nm and calculated the content of Evans blue based on the standard curve.

## Immunofluorescence staining

Frozen slices were roasted at 37°C for 30 minutes, then rinsed with PBS. sealed in a 37°C wet box with 2% BSA or 10% BSA for 30 minutes. Added appropriately diluted mouse anti NeuN (1:200, Abcam UK) to the sample slices, and diluted the first anti myeloperoxidase (MPO) (1:200, Beyotime, China) with PBS with a ratio of 1:200. Until the tissue was completely covered, and placed it overnight in a wet box at 4°C. Then, added Cy3 labeled IgG goat anti rabbit (1:200, Beyotime, China) secondary antibody, incubated in darkness at 37°C for 60 minutes, and rinsed three times with PBS. On the basis of the instructions of the Tunel cell apoptosis detection kit, prepared and dropped the reaction solution. Incubated in the dark at 37°C for 60 minutes, rinsed with PBS three times, stained with DAPI for 3 minutes, and sealed. Selected more than five slices from each group and observed them with fluorescence microscope. All images were captured by fluorescence microscope (Olympus, Tokyo, Japan). Analyzed the fluorescence intensity and number of neurons with Image Pro Plus software.

## Statistical analysis

SPSS 22.0 computer software (SPSS, USA) was used to analyzed the data. All data were presented as Mean ± SD. Analyze multiple comparison procedures with one-way ANOVA. P-value <0.05 indicated statistical significance.

# Results

## miR-199a-5p mimic improved behavioral in MCAO rats

After 24 hours of cerebral ischemia and miRNA treatment, neurobehavior of MCAO rats were assessed by LB 12-Scoring. The results suggested that the LB 12-Scoring of MCAO rats were significantly reduced (P<0.05). Cerebral ischemic caused serious neurological lesion, which

(Data of Ludmila Belayev 12-point scoring between groups)

| | Sham | | | | | MCAO | | | | | miRNA-NC | | | | | miR-199a-5p | | | | |
|---|---|---|---|---|---|---|---|---|---|---|---|---|---|---|---|---|---|---|---|---|
| LB-12 point | 0 | 0 | 0 | 0 | 0 | 10 | 11 | 9 | 9 | 8 | 9 | 11 | 11 | 10 | 9 | 8 | 6 | 7 | 7 | 6 |

**Fig 1. Ludmila Belayev 12-point scoring between groups.**

was obviously improved by miR-199a-5p mimic treatment. however, the rats in miR-NC group show opposite outcomes (Fig 1).

## MiR-199a-5p increased the mRNA expression of Claudin-5 and VEGF in the infarcted area and penumbra, reduced the lesion to the BBB of MCAO rats

RT-PCR was used for assessing the expression of miR-199a-5p, Claudin-5, and VEGF in serum, ischemic penumbra, and infarcted area. The results indicated that the expression level of miR-199a-5p and Claudin-5 were remarkably enhanced in serum, ischemic penumbra, and infarcted area (Fig 2A and 2B), at 24 hours after MCAO. MiR-199a-5p mimic treatment alleviated the lesion of cerebral ischemia to the BBB. Except for the Sham group, there was no noticeable difference in the expression of VEGF in the serum of MCAO rats among the other groups. The expression of VEGF in the Sham group was higher than those of cerebral ischemic rats (Fig 2C). The miR-199a-5p mimic obviously increased the expression of VEGF in the infarcted area and penumbra. Nevertheless, the expression of Claudin-5 and VEGF in the infarcted area and penumbra of miR-NC group were lower than MiR-199a-5p group (Fig 2A–2C).

## MiR-199a-5p protected the integrity of the BBB through regulating Akt signaling pathway

Western blotting was used to explore whether the miR-199a-5p maintained the integrity of BBB by regulating Akt signaling pathway. The expression levels of Claudin-5, VEGF, PIK3R2,

(Data of RT-PCR)

Data of Claudin-5 mRNA expression

| | Sham | | | | | MCAO | | | | | miRNA-NC | | | | | miR-199a-5p | | | | |
|---|---|---|---|---|---|---|---|---|---|---|---|---|---|---|---|---|---|---|---|---|
| Serum | 0.301 | 0.292 | 0.300 | 0.293 | 0.303 | 0.314 | 0.323 | 0.318 | 0.322 | 0.329 | 0.292 | 0.304 | 0.299 | 0.322 | 0.299 | 0.303 | 0.341 | 0.312 | 0.325 | 0.338 |
| infarcted area | 0.298 | 0.296 | 0.301 | 0.291 | 0.301 | 0.590 | 0.626 | 0.616 | 0.671 | 0.633 | 0.560 | 0.554 | 0.591 | 0.544 | 0.532 | 0.796 | 0.764 | 0.813 | 0.837 | 0.826 |
| penumbra | 0.296 | 0.301 | 0.299 | 0.295 | 0.298 | 0.799 | 0.811 | 0.817 | 0.833 | 0.854 | 0.676 | 0.707 | 0.688 | 0.675 | 0.684 | 1.182 | 1.186 | 1.207 | 1.226 | 1.216 |

Data of VEGF-A mRNA expression

| | Sham | | | | | MCAO | | | | | miRNA-NC | | | | | miR-199a-5p | | | | |
|---|---|---|---|---|---|---|---|---|---|---|---|---|---|---|---|---|---|---|---|---|
| Serum | 0.502 | 0.501 | 0.505 | 0.500 | 0.503 | 0.514 | 0.523 | 0.518 | 0.522 | 0.529 | 0.468 | 0.471 | 0.469 | 0.473 | 0.462 | 0.540 | 0.534 | 0.531 | 0.525 | 0.541 |
| infarcted area | 0.499 | 0.496 | 0.501 | 0.491 | 0.493 | 0.469 | 0.472 | 0.466 | 0.471 | 0.463 | 0.356 | 0.361 | 0.359 | 0.354 | 0.349 | 0.696 | 0.714 | 0.713 | 0.738 | 0.722 |
| penumbra | 0.495 | 0.495 | 0.499 | 0.496 | 0.498 | 0.590 | 0.588 | 0.581 | 0.583 | 0.585 | 0.476 | 0.472 | 0.481 | 0.475 | 0.484 | 0.932 | 0.886 | 0.892 | 0.896 | 0.916 |

Data of miR-199a-5p mRNA expression

| | Sham | | | | | MCAO | | | | | miRNA-NC | | | | | miR-199a-5p | | | | |
|---|---|---|---|---|---|---|---|---|---|---|---|---|---|---|---|---|---|---|---|---|
| Serum | 0.322 | 0.311 | 0.305 | 0.330 | 0.316 | 0.384 | 0.383 | 0.358 | 0.379 | 0.366 | 0.358 | 0.376 | 0.369 | 0.352 | 0.362 | 0.380 | 0.364 | 0.373 | 0.385 | 0.367 |
| infarcted area | 0.312 | 0.306 | 0.298 | 0.312 | 0.301 | 0.462 | 0.457 | 0.444 | 0.465 | 0.428 | 0.356 | 0.361 | 0.370 | 0.354 | 0.364 | 0.652 | 0.644 | 0.663 | 0.680 | 0.677 |
| penumbra | 0.318 | 0.316 | 0.317 | 0.314 | 0.318 | 0.590 | 0.570 | 0.581 | 0.593 | 0.585 | 0.456 | 0.472 | 0.461 | 0.465 | 0.454 | 0.782 | 0.806 | 0.812 | 0.806 | 0.816 |

**Fig 2. Expression level of MiR-199a-5p, Claudin-5 mRNA, VEGF-A mRNA, in Serum, penumbra, and infarcted area between groups.**

(Data of western-blotting )

Data of Claudin-5、VEGF-A、PIK3R2 and P-AKt/Akt expression

| | Sham | | | | | MCAO | | | | | miRNA-NC | | | | | miR-199a-5p | | | | |
|---|---|---|---|---|---|---|---|---|---|---|---|---|---|---|---|---|---|---|---|---|
| Claudin-5 | 0.282 | 0.301 | 0.295 | 0.290 | 0.303 | 0.474 | 0.483 | 0.488 | 0.472 | 0.479 | 0.488 | 0.491 | 0.489 | 0.483 | 0.479 | 0.780 | 0.764 | 0.771 | 0.765 | 0.781 |
| VEGF-A | 0.799 | 0.806 | 0.801 | 0.791 | 0.793 | 0.469 | 0.472 | 0.456 | 0.470 | 0.451 | 0.456 | 0.441 | 0.439 | 0.454 | 0.449 | 0.916 | 0.924 | 0.913 | 0.938 | 0.922 |
| PIK3R2 | 0.291 | 0.276 | 0.289 | 0.286 | 0.271 | 0.526 | 0.500 | 0.511 | 0.529 | 0.518 | 0.786 | 0.792 | 0.781 | 0.779 | 0.784 | 0.302 | 0.316 | 0.304 | 0.318 | 0.300 |
| P-AKt/Akt | 0.892 | 0.906 | 0.902 | 0.896 | 0.909 | 0.432 | 0.426 | 0.442 | 0.446 | 0.433 | 0.407 | 0.428 | 0.417 | 0.423 | 0.414 | 0.686 | 0.670 | 0.668 | 0.684 | 0.667 |

**Fig 3. Western blots and semiquantitative analysis of Claudin-5, VEGF-A, PIK3R2, P-Akt/Akt in cerebral ischemic penumbra between groups.**

p-Akt, and Akt were measured by Western blotting (Fig 3). The expression level of PIK3R2 was significantly increased following cerebral ischemic. Compared with miR-NC group, miR-199a-5p mimic treatment obviously reduced PIK3R2 expression (Fig 3C) and raised Claudin-5, VEGF expression (Fig 3A and 3B), and protected the integrity of BBB. Detecting the activation level of Akt in the penumbra area of cerebral ischemia, miR-199a-5p mimic markedly reversed the Akt inactivation caused by cerebral ischemia (Fig 3D). The data suggested that miR-199a-5p mimic protected the integrity of BBB following ischemic stroke.

## MiR-199a-5p decreased BBB permeability after cerebral ischemia in rats

The Evans blue penetration test was used to evaluate the permeability of BBB in rats (Fig 4). The exudation of Evans blue in the MCAO group was higher than that in the Sham group, that indicated the BBB permeability was enhanced after ischemia stroke. Compared with the MCAO group and miR-NC group, miR-199a-5p mimic clearly reduced cerebral vascular permeability after MCAO (Fig 4B). These results suggested that miR-199a-5p mimic exerted a protective effect on BBB integrity after cerebral ischemia.

## MiR-199a-5p inhibited leukocyte infiltration after MCAO

In order to investigate whether miR-199a-5p reduced leukocyte cells infiltration in the penumbra of MCAO rats, we detected the number of MPO+ cells using immunofluorescence staining (Fig 5). The outcomes indicated that the number of MPO+ cells in MCAO rats was dramatically boosted. Following miR-199a-5p mimic treatment, the number of MPO+ cells was remarkablely decreased, while the number of MPO+ cells in the miR-NC group was obviously increased. The results suggested that miR-199a-5p could inhibit leukocyte cells infiltration after MCAO (Fig 5A and 5B).

## MiR-199a-5p lessened apoptosis in the ischemic penumbra of MCAO rats

Immunohistochemistry was used to detect the number of TUNEL+ cells in the ischemic penumbra of MCAO rats (Fig 6). Compared with the MCAO group and miR-NC group, the number of TUNEL+ cells in the miR-199a-5p group was significantly reduced, indicated that miR-199a-5p decreased apoptosis in the penumbra of cerebral ischemia (Fig 6A and 6B).

(Data of Evans Blue extravasation (μg/g tissue) )

| | Sham | | | | | MCAO | | | | | miRNA-NC | | | | | miR-199a-5p | | | | |
|---|---|---|---|---|---|---|---|---|---|---|---|---|---|---|---|---|---|---|---|---|
| Evans Blue extravasation | 2.220 | 2.390 | 2.291 | 2.373 | 2.260 | 7.433 | 7.121 | 7.942 | 7.288 | 7.562 | 8.828 | 8.407 | 8.412 | 8.682 | 8.566 | 3.676 | 3.387 | 3.568 | 3.633 | 3.499 |

**Fig 4. Evans Blue extravasation between groups.**

(Data of MPO+ cell number/mm$^2$)

| | Sham | | | | | MCAO | | | | | miRNA-NC | | | | | miR-199a-5p | | | | |
|---|---|---|---|---|---|---|---|---|---|---|---|---|---|---|---|---|---|---|---|---|
| MPO+ cell number/mm$^2$ | 38 | 37 | 40 | 40 | 39 | 342 | 330 | 338 | 336 | 340 | 371 | 380 | 370 | 379 | 378 | 213 | 229 | 219 | 220 | 223 |

**Fig 5. Quantification data of the MPO+ cells in cerebral ischemic penumbra between groups.**

(Data of Apoptosis cell rate(%)（Tunel）)

| | Sham | | | | | MCAO | | | | | miRNA-NC | | | | | miR-199a-5p | | | | |
|---|---|---|---|---|---|---|---|---|---|---|---|---|---|---|---|---|---|---|---|---|
| Apoptosis cell rate | 6.121 | 8.011 | 10.210 | 6.644 | 7.834 | 36.342 | 38.761 | 42.268 | 39.336 | 40.299 | 44.446 | 47.160 | 43.820 | 47.326 | 46.318 | 23.982 | 31.344 | 28.912 | 26.766 | 30.855 |

**Fig 6. The rate of apoptosis cell in cerebral ischemic penumbra between groups.**

(Data of NeuN+/TUNEL+ cells number/mm$^2$)

| | Sham | | | | | MCAO | | | | | miRNA-NC | | | | | miR-199a-5p | | | | |
|---|---|---|---|---|---|---|---|---|---|---|---|---|---|---|---|---|---|---|---|---|
| NeuN+/TUNEL+ cells number/mm2 | 6 | 3 | 3 | 4 | 5 | 37 | 42 | 40 | 42 | 39 | 52 | 50 | 47 | 48 | 50 | 19 | 22 | 24 | 23 | 21 |

**Fig 7. The rate of apoptosis neurons in cerebral ischemic penumbra between groups.**

## MiR-199a-5p inhibited neuronal apoptosis in the ischemic penumbra of MCAO rats

NeuN+/TUNEL+ double stained cells were investigated by immunofluorescence (Fig 7). The results suggested that there were almost no apoptotic neurons in the Sham group. MCAO rats showed a large number of neuronal apoptosis in the ischemic penumbra. However, miR-199a-5p mimic significantly diminished neuronal apoptosis (Fig 7A and 7B).

## Discussion

Blood brain barrier injury was an important pathological process in the pathogenesis of ischemic stroke, which probably disrupted the homeostasis of the brain environment, exacerbated inflammatory reactions, and caused neuronal apoptosis. The results of this study showed that after 24 hours of treatment with miR-199a-5p mimic, the limb sensory motor integration ability of MCAO rats was better than that of the MCAO group and miR-NC group. MiR-199a-5p reduced the expression of PIK3R2 protein in MCAO rats, enhanced Akt activation, obviously upregulated the gene and protein expression of Claudin-5 and VEGF in the infarcted area and ischemic penumbra, reduced BBB damage, and protected its integrity. Evans blue staining results showed that compared to the MCAO group and miR-NC group, miR-199a-5p significantly reduced the blood-brain barrier permeability in MCAO rats. Meanwhile, immunofluorescence staining results showed that miR-199a-5p inhibited leukocyte cells infiltration after cerebral ischemia, alleviated inflammatory, and diclined neuronal apoptosis in the ischemic penumbra area.

Neurological deficit score, improved Bederson score, and Longa 5 points method were commonly used for evaluating neurological deficits, but, they didn't provide a detailed evaluation of the overall changes in neurological function. The Ludmila Belayev 12 point scoring method in this study comprehensively assessed the sensory motor integration ability of MCAO rats, and evaluate the neural function of MCAO rats through the combination of visual, tactile, proprioceptive, and motor functions, which were more convincing.

PI3K/AKT was a classic neuroprotective signaling pathway, and its mechanism of protecting neuronal cell survival mainly included: 1) inhibiting neuronal cell apoptosis by producing nerve growth factors and neurotrophic factors [29]; 2) Phosphorylation activated downstream substrates to inhibit cell apoptosis [30]; 3) Intervention of neuronal apoptosis through mitochondrial pathways [31]; 4) Protecting neurons through the erythropoietin and its receptor system. PIK3R2 was a member of the PI3K subunit family, which inhibited the activation of the PI3K/Akt pathway [32–34], regulated cell proliferation, migration, and maturation. Meanwhile, it was also a negative regulatory factor of VEGF [33–35]. MiR-199a-5p alleviated BBB damage, suppressed inflammatory reaction, and neuron damage caused by cerebral ischemia by inhibiting PIK3R2 and activating Akt.

BBB played a crucial role in regulating brain metabolism and maintaining central nervous system homeostasis. TJ proteins were the basis structural of BBB [13,15,36]. Claudin-5 was a key TJ protein that maintained the integrity and permeability of BBB, and its damage was the beginning of the disruption of BBB integrity in many pathological processes of brain injury diseases [15,37–39]. Vascular endothelial cells were a prominent structure that constituted the BBB, VEGF was a key regulatory factor for endothelial growth. Promoting the binding of VEGF to its receptors was great significance for protecting the integrity of BBB [40–42]. 24 hours after cerebral ischemia in rats, it was the peak period of brain edema and inflammatory reaction [24,43]. The expression level of TJ protein Claudin-5 was evidently diminished [6,35,44], and the permeability of BBB was increased. MicroRNA has been proven to alleviate inflammation and edema after cerebral ischemia, promote nerve regeneration and angiogenesis [33,45–47]. MiR-199a-5p intervention promoted the expression of Claudin-5 and VEGF, protect the integrity of BBB, and alleviated local inflammatory. Researches had found that miR-199a-5p was involved in regulating neuronal regeneration in the hippocampus and subventricular area after cerebral ischemia, and alleviated edema caused by cerebral ischemia. The results of this study suggested that miR-199a-5p cut down neuronal apoptosis in the ischemic penumbra, which could be closely related to the involvement of miR-199a-5p in reducing BBB permeability and alleviating local inflammatory. Other studies had shown that the level of Claudin-5 decreased in the early stages of vascular remodeling and increased in the later stages, and this dynamic change was closely related to the dynamic changes in BBB permeability [22,46–48]. In the present study, miR-199a-5p upregulated the expression of Claudin-5 and VEGF by regulating the PI3K/AKT signaling pathway, promoting endothelial neurogenesis in the ischemic penumbra and reducing the permeability of Evans blue.

Myeloperoxidase (MPO) was a functional and activation marker of neutrophils, a vital marker of inflammatory, and closely related to the inflammatory after cerebral ischemia [5,8]. MPO mediated high-density lipoprotein oxidation weaken endothelial cell proliferation and migration, and inhibited the activation of the Akt signaling pathway [26]. Bushueva et al. [27] found that the CpG site of the MPO gene in leukocyte cells of stroke patients undergone significant hypomethylation, exacerbating oxidative stress. MiR-199a-5p downregulated the expression of PIK3R2 and activated the Akt signaling pathway, alleviating inflammatory cell infiltration in the ischemic penumbra of MCAO rats.

Although the present study has achieved some meaningful results, there are still some limitations. Include the volume of cerebral infarction was not assessed in MCAO rats, and the brain water content was not measured before and after treatment in MCAO rats.

## Conclusion

In summary, the findings of the present study indicated that miR-199a-5p inhibited inflammation following ischemic stroke by activating the PI3K/Akt signaling pathway, reduced

neuronal apoptosis in the ischemic penumbra, and upregulated Claudin-5 and VEGF to preserve the integrity of the BBB. miR-199a-5p will be anticipated to serve as a novel therapeutic target for cerebral ischemia. However, this study still has limitations. While exploring the mechanism by which miR-199a-5p protected the BBB integrity after cerebral ischemia using miR-199a-5p mimics and negative controls, the role of endogenous miR-199a-5p cannot be fully ruled out.

In the future, the research team will investigate the mechanisms by which miR-199a-5p preserves BBB integrity following cerebral ischemia, either through silencing or using PI3K/Akt signaling pathway blockers, and the mechanisms by which miR-199a-5p promotes neurogenesis following ischemic stroke.

## Author Contributions

**Data curation:** Lulu Kou, Ran Meng.

**Funding acquisition:** Ran Meng.

**Methodology:** Chunqiao Duan, Ran Meng.

**Project administration:** Guangxiao Ni.

**Software:** Lulu Kou.

**Writing – original draft:** Chunqiao Duan, Pu Wang.

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
