## [Decision Letter · Decision Letter 0]

2 Apr 2024

PONE-D-24-00772MicroRNA-199a-5p attenuates blood-brain barrier disruption following ischemic stroke by regulating PI3K/Akt signaling pathwayPLOS ONE

Dear Dr. NI,

Thank you for submitting your manuscript to PLOS ONE. After careful consideration, we feel that it has merit but does not fully meet PLOS ONE’s publication criteria as it currently stands. Therefore, we invite you to submit a revised version of the manuscript that addresses the points raised during the review process.

We look forward to receiving your revised manuscript.

Kind regards,

Francesco Sessa, Ph.D., MS

Academic Editor

PLOS ONE

Journal Requirements:

"National Natural Science Foundation of China，（No.81273609）

Natural Science Foundation of Hebei Province，(No. H2023206115)"

5. Please ensure that you include a title page within your main document. You should list all authors and all affiliations as per our author instructions and clearly indicate the corresponding author.

7. Please include a separate caption for each figure in your manuscript.

8. Please include your tables as part of your main manuscript and remove the individual files. Please note that supplementary tables (should remain/ be uploaded) as separate ""supporting information"" files

**Additional Editor Comments:**

The reviewers raised several important concerns. I believe that authors should further improve their manuscript. Please, revise the manuscript solving all criticisms and providing the rebuttal letter.

Reviewers' comments:

Reviewer's Responses to Questions

**Comments to the Author**

1. Is the manuscript technically sound, and do the data support the conclusions?

Reviewer #1: No

Reviewer #2: Yes

2. Has the statistical analysis been performed appropriately and rigorously? 

Reviewer #1: Yes

Reviewer #2: Yes

3. Have the authors made all data underlying the findings in their manuscript fully available?

Reviewer #1: No

Reviewer #2: Yes

4. Is the manuscript presented in an intelligible fashion and written in standard English?

Reviewer #1: No

Reviewer #2: No

5. Review Comments to the Author

Reviewer #1: In this manuscript, the authors studied the preventive and therapeutic effects of miR-199a-5p through PI3K/Akt pathway on ischemia stroke. On this basis, they further investigated the role of miR-199a-5p in BBB. Despite some highlights, this paper has a number of shortcomings, such as the logic of manuscript, the data etc, and page numbers and line numbers are missing.

1.The introduction lacks logic. Especially, it did not describe the relationship between miRNA and BBB. Please describe in detail.

2.Where are the tables? I can’t find them. Neurological function scores are better presented as bar graphs than table.

3.The improvement effect of miRNA on MCAO was only evaluated with neurological function score. It should add the data of infarct size and pathological status.

4.Although this study provides detailed evidence of the efficacy of CC, it does not provide in-depth mechanistic validation.

5.There is no logic in the results section. It should be showed from effect to mechanism.

6.Fig. 6A should add the merged figure.

7.The discussion lacks logic. Moreover, miR-199a-5p, as the key to this paper, has not been highlighted in the discussion.

8.Another obvious problem with this paper is lack of sufficient explanation of the results. More explanations on them seem necessary and helpful to readers.

9.The first paragraph of the discussion requires a brief summary of the research results.

10. It is noted that your manuscript needs careful editing by someone with expertise in technical English editing paying particular attention to English grammar, spelling, and sentence structure so that the goals and results of the study are clear to the reader.

Reviewer #2: This is a well-organized study exploring the association of MicroRNA-199a-5p attenuates BBB disruption following ischemic stroke.

Major:

The concept of the study is not novel and the topic itself is important, however, the lack of clear rationale and mechanistic assessments reduces the scientific value of the paper.

The English used in the manuscript is understandable but the text needs to be edited by a native speaker. Spelling, grammar, and syntax errors are present in the text. In the study implications, novelty aspects of the study and limitations are not indicated.

Minor:

please correct abbreviations - for example, in the introduction, CNS has already been used before, and TJs, not TJS

- access to food and not diet

- there are some misleading sentences in the text for example that miR-199a-5p was a main factor in the process of cerebral ischemia - introduction

6. PLOS authors have the option to publish the peer review history of their article (what does this mean?). If published, this will include your full peer review and any attached files.

Reviewer #1: No

Reviewer #2: No

---

## [Author Response · Author response to Decision Letter 0]

28 May 2024

Response to Reviewers

Dear Editors and Reviewers:

 Thank you for your letter and for the reviewers’ comments concerning our manuscript entitled “MicroRNA-199a-5p attenuates blood-brain barrier disruption following ischemic stroke by regulating PI3K/Akt signaling pathway”. (ID: PONE-D-24-00772). Those comments are all valuable and very helpful for revising and improving our paper, as well as the important guiding significance to the research. We have studied comments carefully and have made correction which we hope meet with approval. Revised portion are marked in red in the paper. The main corrections in the paper and the responds to the reviewer’s comments are as following: Responds to the reviewer’s comments:

1.A marked-up copy of your manuscript that highlights changes made to the original version. You should upload this as a separate file labeled 'Revised Manuscript with Track Changes'. 

Response: A separate file labeled 'Revised Manuscript with Track Changes' has been uploaded. 

2.Please state what role the funders took in the study. If the funders had no role, please state: ""The funders had no role in study design, data collection and analysis, decision to publish, or preparation of the manuscript." 

Response : The funders "National Natural Science Foundation of China（No.81273609)" had a role in study design and preparation of the manuscript. The funders " Natural Science Foundation of Hebei Province (No. H2023206115)" had a role in data collection and analysis, decision to publish. 

4. For example, authors should submit the following data: The values behind the means, standard deviations and other measures reported; The values used to build graphs; The points extracted from images for analysis. Authors do not need to submit their entire data set if only a portion of the data was used in the reported study.

Response : The data has been upload as a separate file labeled " date of graphs and figures" has been uploaded. 

Reviewer #1: In this manuscript, the authors studied the preventive and therapeutic effects of miR-199a-5p through PI3K/Akt pathway on ischemia stroke. On this basis, they further investigated the role of miR-199a-5p in BBB. Despite some highlights, this paper has a number of shortcomings, such as the logic of manuscript, the data etc, and page numbers and line numbers are missing.

Response: Considering the Reviewer’s suggestion, we have added page numbers and line numbers.

1.The introduction lacks logic. Especially, it did not describe the relationship between miRNA and BBB. Please describe in detail.

Response: thank you for your suggestion. We have revised the introduction and described the relationship between miRNA and BBB.

2.Where are the tables? I can’t find them. Neurological function scores are better presented as bar graphs than table.

Response: The tables were uploaded as separate file. Neurological function scores have been presented as bar graphs.

3.The improvement effect of miRNA on MCAO was only evaluated with neurological function score. It should add the data of infarct size and pathological status.

Response: thank you for your suggestion. In the present study, we primarily investigated the protective effect of miR-199a-5p on the BBB in MCAO rats through regulating PI3K/Akt signaling pathway. Thus, the measurement of cerebral infarction volume was not considered in the study design.

4.Although this study provides detailed evidence of the efficacy of CC, it does not provide in-depth mechanistic validation.

Response: In our study, the efficacy of CC was not validated, and it was not mentioned in the manuscript. 

5.There is no logic in the results section. It should be showed from effect to mechanism.

Response: In the results section, the results of the experiment were presented, and the relevant mechanisms are presented in the discussion section.

6.Fig. 6A should add the merged figure.

Response: In Fig. 6A, the left side showed the staining of neuronal nuclei, and the right side showed the result of NEUN+/TUNEL+ staining, which was the merge figure. 

7.The discussion lacks logic. Moreover, miR-199a-5p, as the key to this paper, has not been highlighted in the discussion.

Response: Thank you for the reviewer's comments. As per the reviewer's recommendations, the discussion has been revised. Special emphasis was placed on miR-199a-5p.

8.Another obvious problem with this paper is lack of sufficient explanation of the results. More explanations on them seem necessary and helpful to readers.

Response: Thank you for the reviewer's suggestion. In the discussion section, a detailed explanation of the results was provided.

9.The first paragraph of the discussion requires a brief summary of the research results.

Response: Thank you for the reviewer's comments. A brief summary of the research results was presented in the first paragraph of the discussion section.

10. It is noted that your manuscript needs careful editing by someone with expertise in technical English editing paying particular attention to English grammar, spelling, and sentence structure so that the goals and results of the study are clear to the reader.

Response: We appreciate the reviewer's suggestions. The manuscript has been revised by professionals to correct the grammar, tense, and sentence structure in the text, making it easier to read.

Reviewer #2: This is a well-organized study exploring the association of MicroRNA-199a-5p attenuates BBB disruption following ischemic stroke.

Major:

The concept of the study is not novel and the topic itself is important, however, the lack of clear rationale and mechanistic assessments reduces the scientific value of the paper.

The English used in the manuscript is understandable but the text needs to be edited by a native speaker. Spelling, grammar, and syntax errors are present in the text. In the study implications, novelty aspects of the study and limitations are not indicated.

Response: We appreciate the reviewer's suggestions. The findings of the present study indicated that miR-199a-5p inhibit inflammation following ischemic stroke by activating the PI3K/Akt signaling pathway, reduced neuronal apoptosis in the ischemic penumbra, and upregulated Claudin-5 and VEGF to preserve the integrity of the BBB. miR-199a-5p will be anticipated to serve as a novel therapeutic target for cerebral ischemia.

The manuscript has been revised by professionals to correct the grammar, tense, and sentence structure in the text, making it easier to read.

Reviewer #2: This is a well-organized study exploring the association of MicroRNA-199a-5p attenuates BBB disruption following ischemic stroke.

Minor: 

please correct abbreviations - for example, in the introduction, CNS has already been used before, and TJs, not TJS

- access to food and not diet

-there are some misleading sentences in the text for example that miR-199a-5p was a main factor in the process of cerebral ischemia - introduction

Response: We appreciate the reviewer's attention to detail. We have made revisions one by one according to the suggestions.

---

## [Decision Letter · Decision Letter 1]

18 Jun 2024

PONE-D-24-00772R1MicroRNA-199a-5p attenuates blood-brain barrier disruption following ischemic stroke by regulating PI3K/Akt signaling pathwayPLOS ONE

Dear Dr. NI,

Thank you for submitting your manuscript to PLOS ONE. After careful consideration, we feel that it has merit but does not fully meet PLOS ONE’s publication criteria as it currently stands. Therefore, we invite you to submit a revised version of the manuscript that addresses the points raised during the review process.

**ACADEMIC EDITOR: **Despite Reviewer#1 rejecting the manuscript, in my opinion, the authors improved the manuscript substantially. I suggest revisiting it and inserting a separate section with the study limitations (the authors should insert the limitations described by reviewer#1). Please, insert it before conclusion: in this way, it could be published.

We look forward to receiving your revised manuscript.

Kind regards,

Francesco Sessa, Ph.D., MS

Academic Editor

PLOS ONE

Journal Requirements:

**Additional Editor Comments:**

Despite Reviewer#1 rejecting the manuscript, in my opinion, the authors improved the manuscript substantially. I suggest revisiting it and inserting a separate section with the study limitations (the authors should insert the limitations described by reviewer#1: "Although the authors have made numerous modifications for the manuscript, the current data do not support the conclusions. The data of infarct size and pathological status are important index for evaluating ischemic stroke. They are not provided.").

Reviewers' comments:

Reviewer's Responses to Questions

**Comments to the Author**

1. If the authors have adequately addressed your comments raised in a previous round of review and you feel that this manuscript is now acceptable for publication, you may indicate that here to bypass the “Comments to the Author” section, enter your conflict of interest statement in the “Confidential to Editor” section, and submit your "Accept" recommendation.

Reviewer #1: (No Response)

Reviewer #2: All comments have been addressed

2. Is the manuscript technically sound, and do the data support the conclusions?

Reviewer #1: Partly

Reviewer #2: Yes

3. Has the statistical analysis been performed appropriately and rigorously? 

Reviewer #1: I Don't Know

Reviewer #2: Yes

4. Have the authors made all data underlying the findings in their manuscript fully available?

Reviewer #1: No

Reviewer #2: Yes

5. Is the manuscript presented in an intelligible fashion and written in standard English?

Reviewer #1: Yes

Reviewer #2: Yes

6. Review Comments to the Author

Reviewer #1: Although the authors have made numerous modifications for the manuscript, the current data do not support the conclusions. The data of infarct size and pathological status are important index for evaluating ischemic stroke. They are not provided.

Reviewer #2: (No Response)

7. PLOS authors have the option to publish the peer review history of their article (what does this mean?). If published, this will include your full peer review and any attached files.

Reviewer #1: No

Reviewer #2: No

---

## [Author Response · Author response to Decision Letter 1]

21 Jun 2024

The study limitations have been inserted before conclusion.Xiaoqing Wang was listed in the authorship.

---

## [Editor Report · Decision Letter 2]

25 Jun 2024

MicroRNA-199a-5p attenuates blood-brain barrier disruption following ischemic stroke by regulating PI3K/Akt signaling pathway

PONE-D-24-00772R2

Dear Dr. NI,

We’re pleased to inform you that your manuscript has been judged scientifically suitable for publication and will be formally accepted for publication once it meets all outstanding technical requirements.

Kind regards,

Francesco Sessa, Ph.D., MS

Academic Editor

PLOS ONE

Additional Editor Comments (optional):

The authors modified the manuscript following the reviewers' suggestions.
---

## [Editor Report · Acceptance letter]

27 Jun 2024

PONE-D-24-00772R2 

PLOS ONE

Dear Dr. NI, 

I'm pleased to inform you that your manuscript has been deemed suitable for publication in PLOS ONE. Congratulations! Your manuscript is now being handed over to our production team.

Kind regards, 

on behalf of

Lecturer Francesco Sessa 

Academic Editor

PLOS ONE